



# Global distribution of methane emissions: a comparative inverse analysis of observations from the TROPOMI and GOSAT satellite instruments

Zhen Qu[1], Daniel J. Jacob[1], Lu Shen[1], Xiao Lu[1], Yuzhong Zhang[1,2], Tia R. Scarpelli[1], Hannah O. Nesser[1],
5 Melissa P. Sulprizio[1], Joannes D. Maasakkers[3], A. Anthony Bloom[4], John R. Worden[4], Robert J. Parker[5,6],
Alba L. Delgado[3]

[1]School of Engineering and Applied Science, Harvard University, Cambridge, MA, USA
[2]Key Laboratory of Coastal Environment and Resources of Zhejiang Province (KLaCER), School of Engineering, Westlake
University, Hangzhou, Zhejiang, China
10 [3]SRON Netherlands Institute for Space Research, Utrecht, the Netherlands
[4]Jet Propulsion Laboratory, California Institute of Technology, Pasadena, CA, USA
[5]National Centre for Earth Observation, University of Leicester, Leicester, UK
[6]Earth Observation Science, School of Physics and Astronomy, University of Leicester, UK

*Correspondence to*: Zhen Qu (zhenqu@g.harvard.edu)

**Abstract.** We evaluate the global atmospheric methane column retrievals from the new TROPOMI satellite instrument and apply them to a global inversion of methane sources for 2019 at 2°×2.5° horizontal resolution. We compare the results to an inversion using the sparser but more mature GOSAT satellite retrievals, as well as a joint inversion using both TROPOMI and 20 GOSAT. Validation of TROPOMI and GOSAT with TCCON ground-based measurements of methane columns, after correcting for retrieval differences in prior vertical profiles and averaging kernels using the GEOS-Chem chemical transport model, shows global biases of -2.7 ppbv for TROPOMI and -1.0 ppbv for GOSAT, and regional biases of 6.7 ppbv for TROPOMI and 2.9 ppbv for GOSAT. Intercomparison of TROPOMI and GOSAT shows larger regional discrepancies exceeding 20 ppbv, mostly over regions with low surface albedo in the shortwave infrared where the TROPOMI retrieval may 25 be biased. Our inversion uses an analytical solution to the Bayesian optimization of methane sources, thus providing an explicit characterization of error statistics and information content together with the solution. TROPOMI has ~100 times more observations than GOSAT but error correlation on the 2°×2.5° scale of the inversion and large spatial variations of the number of observations make it less useful than GOSAT for quantifying emissions at that resolution. Finer-scale regional inversions would take better advantage of the TROPOMI data density. The TROPOMI and GOSAT inversions show consistent downward 30 adjustments of global oil/gas emissions relative to a prior estimate based on national inventory reports to the United Nations Framework Convention on Climate Change, but consistent increases in the south-central US and in Venezuela. Global emissions from livestock (the largest anthropogenic source) are adjusted upward by TROPOMI and GOSAT relative to the EDGAR v4.3.2 prior estimate. We find large artifacts in the TROPOMI inversion over Southeast China, where seasonal rice



emissions are particularly high but in phase with extensive cloudiness, and where coal emissions may be misallocated. Future advances in the TROPOMI retrieval together with finer-scale inversions and improved accounting of error correlations should enable improved exploitation of TROPOMI observations to quantify and attribute methane emissions on the global scale.

# 1 Introduction

Methane ($CH_4$) is the second most important anthropogenic greenhouse gas in the atmosphere after $CO_2$. It is emitted to the atmosphere naturally, mainly from wetlands. Anthropogenic sources include the oil/gas industry, coal mining, livestock, rice agriculture, landfills, and wastewater treatment. Methane loss in the atmosphere is mainly by oxidation by the hydroxyl radical (OH). This oxidation leads to the production of other greenhouse gases (ozone, stratospheric water vapor, and $CO_2$), which together with methane add up to a radiative forcing of 0.97 W m$^{-2}$ since pre-industrial times [Myhre et al., 2013]. Climate action on methane requires quantification of its emissions but current inventories are highly uncertain [Saunois et al., 2020]. Satellite observations of atmospheric methane columns can evaluate and improve these inventories using inverse analyses [Jacob et al., 2016], and this has been extensively done with the GOSAT instrument launched in 2009 [Monteil et al., 2013; Cressot et al., 2014; Alexe et al., 2015; Pandey et al., 2016; Maasakkers et al., 2019; Lu et al., 2021; Y. Zhang et al., 2021]. The TROPOspheric Monitoring Instrument (TROPOMI) launched in October 2017 now provides a much higher observation density than GOSAT [Hu et al., 2018]. Here we present a global inverse analysis of one year (2019) of these early TROPOMI observations to evaluate their capability for quantifying methane emissions, comparing to an inversion for that same year using the sparser but more mature observations from GOSAT.

Both TROPOMI and GOSAT measure atmospheric methane columns by backscatter of solar radiation in the shortwave infrared (SWIR). TROPOMI observes at 2305-2385 nm and retrieves methane columns with a full-physics algorithm [Connor et al., 2008; Butz et al., 2011]. GOSAT observes at 1630-1700 nm, which enables retrieval by the $CO_2$ proxy method taking advantage of $CO_2$ absorption in that same band [Parker et al., 2020a]. The full-physics approach does not depend on prior information on the $CO_2$ column, but the retrieval is more vulnerable to scattering artefacts. Therefore, TROPOMI has very strict filtering and its retrieval success rate is only 3% [Hasenkamp et al., 2019]. GOSAT has a much higher retrieval success rate of 24% limited mainly by cloud cover [Parker et al., 2020a]. The reported precisions of TROPOMI and GOSAT retrievals are comparable, with a value of 0.7% for GOSAT [Kuze et al., 2016; Parker et al., 2020a] and 0.6% for TROPOMI [Butz et al., 2012]. TROPOMI provides continuous daily global coverage with a nadir pixel resolution of $7 \times 7$ km$^2$ ($5.5 \times 7$ km$^2$ after August 2019). GOSAT samples circular pixels of 10.5 km diameter separated by 250 km with a 3-day return time in its standard viewing mode.

Inferring emissions from methane satellite observations requires inversion of a chemical transport model (CTM) that relates emissions to atmospheric concentrations. This is generally done by Bayesian optimization of a posterior emission estimate





given the observations and a prior estimate [Jacob et al., 2016]. Most inverse analyses use a variational solution to the Bayesian optimization problem, which enables inference of emissions at any resolution but does not readily provide error statistics [Meirink et al., 2008; Monteil et al., 2013; Wecht et al., 2014; Stanevich et al., 2019]. Analytical solution has the advantage of including posterior error statistics and hence information content as part of the solution [Brasseur and Jacob, 2017]. It requires

explicit construction of the Jacobian matrix of the CTM but this is readily done with massively parallel computing. Once the Jacobian matrix has been constructed, it can be applied to conduct ensembles of inversions at no added cost exploring the dependence of the solution on inversion parameters or observational data selection. Analytically-based inversions of GOSAT satellite data have been used to pinpoint areas where the inversion results are most informed by the observations [Turner et al., 2015; Maasakkers et al., 2019], to diagnose the ability of the inversion to separate contributions from different source sectors

[Maasakkers et al., 2021; Y. Zhang et al., 2021] and from sources and sinks [Y. Zhang et al., 2018; Maasakkers et al., 2019], and to compare the information content from satellite and suborbital observations [Lu et al., 2021; Baray et al., 2021].

Here we present global analytical inversions of TROPOMI and GOSAT data for 2019 at $2° \times 2.5°$ resolution to infer methane sources and sinks and to attribute emissions to different sectors. This involves evaluation and intercomparison of the

TROPOMI and GOSAT retrievals prior to the inversion, as any biases in the observations will propagate to the inversion results. We compare inversion results for the two instruments separately and jointly. We diagnose the information content of the inversion for each instrument and for the joint system in different regions of the world. This enables us to assess the consistency and complementarity of the two data sets.

## 2 Methane observations

TROPOMI and GOSAT are in Sun-synchronous orbits with local overpass solar times of 13:30 and 13:00 respectively [Veefkind et al., 2012; Kuze et al., 2016]. We use the version 1.03 TROPOMI methane retrieval from the Netherlands Institute for Space Research [Hu et al., 2016] (http://www.tropomi.eu/data-products/methane, last accessed Aug 8, 2020) and the GOSAT methane retrieval version 9.0 of the University of Leicester obtained by the $CO_2$ proxy method [Parker and Boesch, 2020] (https://catalogue.ceda.ac.uk/uuid/18ef8247f52a4cb6a14013f8235cc1eb, last accessed Dec 29, 2020). We use one year

of data (January-December 2019) to optimize methane emissions for 2019. We only include high quality retrievals with "qa_value" $\geq 0.5$ for TROPOMI [S5P MPC, 2020] and "xch4_quality_flag" = 0 for GOSAT. The TROPOMI and GOSAT products are provided as column-averaged dry methane mixing ratios ($X_{CH4}$) along with the prior vertical profiles used in the retrieval procedures and the averaging kernel vectors describing the altitude-dependent sensitivity of the retrievals.

The left panels of Figure 1 show the annual mean $X_{CH4}$ observations from TROPOMI and GOSAT in 2019. We excluded observations poleward of 60° where (1) persistent snow cover leads to low albedo [Hasekamp et al., 2019], (2) low Sun angles and extensive cloud cover make the retrieval more difficult, and (3) stratospheric CTM bias can affect the inversion [Turner et al., 2015]. The TROPOMI retrieval is successful for only 2% of scenes at 60°S – 60°N, still producing 56684576 TROPOMI



observations, which is two orders of magnitude higher than for GOSAT (544911 observations after filtering with the quality
flag). As shown in the right panel of Figure 1, TROPOMI observations are relatively sparse over persistently cloudy regions
such as the wet tropics. GOSAT has relatively more success over these regions because of the use of the $CO_2$ proxy method.
The GOSAT $CH_4$ product also includes observations over the ocean for sunglint geometries and these are not included in the
current TROPOMI product.

We conducted a common evaluation of the TROPOMI and GOSAT observations with ground-based TCCON measurements
of $X_{CH4}$ [TCCON Team, 2017], using the GEOS-Chem CTM to resolve differences in prior estimates and averaging kernels
between the TROPOMI, GOSAT, and TCCON retrievals [L. Zhang et al., 2010]. Only 9 TCCON sites have continuous
observations for the whole year of 2019, but 22 sites have observations over the period of May 2018 – Apr 2019 when
TROPOMI observations started to be available. We therefore focus on the period from May 2018 to April 2019 for evaluation.

Following L. Zhang et al. [2010], we remove the discrepancy from the use of different prior profiles in the TROPOMI, GOSAT,
and TCCON retrievals by substituting a common fixed prior profile, which we take as the annual averaged TROPOMI prior
profile between 30°S and 30°N. This substitution is done only for the purpose of intercomparison; it is not used subsequently
in the inversion. We average the individual retrievals over 2°×2.5° GEOS-Chem grid cells, and apply the averaging kernels of
the individual retrievals to the GEOS-Chem simulated vertical profiles to produce a model simulation of the observations. We
calculate the differences Δ between satellite and TCCON measurements with reference to the GEOS-Chem CTM using:

$$\Delta \,(\text{TROPOMI} - \text{TCCON}) = \left(\hat{X}_{\text{CH4,TROPOMI}} - \hat{X}_{\text{CH4,CTM\_TROPOMI}}\right) - \left(\hat{X}_{\text{CH4,TCCON}} - \hat{X}_{\text{CH4,CTM\_TCCON}}\right), (1)$$

and

$$\Delta \,(\text{GOSAT} - \text{TCCON}) = \left(\hat{X}_{\text{CH4,GOSAT}} - \hat{X}_{\text{CH4,CTM\_GOSAT}}\right) - \left(\hat{X}_{\text{CH4,TCCON}} - \hat{X}_{\text{CH4,CTM\_TCCON}}\right), \qquad (2)$$

where $\hat{X}_{\text{CH4,TROPOMI}}$, $\hat{X}_{\text{CH4,GOSAT}}$, and $\hat{X}_{\text{CH4,TCCON}}$ are methane column mixing ratios from TROPOMI, GOSAT, and TCCON
after substitution with the same prior profile. $\hat{X}_{\text{CH4,CTM\_TROPOMI}}$, $\hat{X}_{\text{CH4,CTM\_GOSAT}}$, and $\hat{X}_{\text{CH4,CTM\_TCCON}}$ are simulated methane
column mixing ratios with the appropriate averaging kernels applied.

Figure 2 shows the mean differences of TROPOMI and GOSAT with TCCON for the 22 TCCON sites. There are large and
correlated differences at the Izana and Zugspitze mountaintop sites where we would not expect consistency with the satellite
data averaged over the 2°×2.5° grid. For the remaining 20 sites, the mean biases are -2.7 ppbv for TROPOMI and -1.0 ppbv
for GOSAT. Of more interest for the inversion are systematic errors on regional scales (regional bias), which can be estimated
by the standard deviations of Δ (TROPOMI – TCCON) and Δ (GOSAT – TCCON) across all TCCON sites [Buchwitz et al.,
2015]. The regional bias diagnoses the reliability of the observed methane gradients for inferring methane sources in the





inversion. We find regional biases of 2.9 ppbv for GOSAT and 6.7 ppbv for TROPOMI. The regional bias for GOSAT is below the "breakthrough requirement" of 5 ppbv set by Buchwitz et al. [2015] as needing to be achieved for regional/global inversions of satellite observations, and the regional bias for TROPOMI is below their "threshold requirement" of 10 ppbv.

This implies that GOSAT observations are of high quality for quantifying methane sources while TROPOMI observations are still useful. The regional bias of GOSAT compared to TCCON is smaller than the value of 3.9 ppbv in Parker et al. [2020a], which may reflect at least in part our accounting for differences in averaging kernels and prior vertical profiles. The larger regional biases from TROPOMI may reflect error correlations of retrieved $X_{CH4}$ and SWIR surface albedo [Hu et al., 2018; Hasekamp et al., 2019; Schneising et al., 2019].


We apply the same method for a more extensive analysis of regional differences between TROPOMI and GOSAT. Figure 3 shows the global distributions of the seasonal mean differences Δ between the two instruments, again correcting for differences in prior estimates and averaging kernels. The seasonal global mean biases for TROPOMI relative to GOSAT are consistent with the comparison to TCCON but the regional biases are larger (8.8-12.8 ppbv), and some regions show differences of

magnitude comparable to the regional enhancements of Figure 1. The regional biases tend to be consistent across seasons, except for positive biases north of 50°N in DJF that could be associated with snow cover. We find particularly large differences between TROPOMI and GOSAT where the SWIR surface albedo is smaller than 0.1 as in Brazil, central Africa, and subarctic regions (see Figure S1). We may therefore expect large differences between TROPOMI and GOSAT inversions for these regions.

**3 Inversion method**

We assemble the 2019 TROPOMI and GOSAT observations of $X_{CH4}$ into an observation vector $\boldsymbol{y}$, and use the observations to optimize a state vector $\boldsymbol{x}$ consisting of methane sources and sinks. We use the GEOS-Chem global CTM version 12.5.0 (10.5281/zenodo.3403111) at 2°×2.5° grid resolution with 47 vertical layers as the forward model in the inversion. Prior estimates $\boldsymbol{x}_a$ for methane sources on that grid are compiled from bottom-up inventories. We solve the Bayesian optimization

problem analytically to obtain both the posterior solution $\hat{\boldsymbol{x}}$ and its error covariance matrix $\hat{\mathbf{S}}$. We conduct inversions using TROPOMI and GOSAT observations separately and together (joint inversion), as well as additional inversions to examine the sensitivity of results to different parameters in the Bayesian optimization.

**3.1 GEOS-Chem simulations and prior estimates**

GEOS-Chem is driven by Modern-Era Retrospective analysis for Research and Applications, Version 2 (MERRA-2)

meteorological fields from the NASA Global Modeling and Assimilation Office (GMAO). The original methane simulation is described by Wecht et al. [2014]. Previous GEOS-Chem-based inversions at 4°×5° horizontal resolution had excessive stratospheric methane poleward of 60° in winter-spring due to the inability to reproduce the polar vortex dynamical barrier,



and this needed to be corrected in the inversion [Turner et al., 2015; Y. Zhang et al., 2021]. The polar vortex dynamics are much better captured at 2°×2.5° resolution [Stanevich et al., 2019; Y. Zhang et al., 2021], and we do not use satellite data

poleward of 60º in our inversion. There is therefore no need for stratospheric bias correction.

Table 1 summarizes the prior estimates of the sources and sinks of methane, and Figure 4 shows the spatial distribution of the sources. The emissions from oil, gas, and coal exploitation are from the 2016 Global Fuel Exploitation Inventory (GFEI) version 1.0 [Scarpelli et al., 2020], which spatially allocates national emissions reported to the United Nations Framework

Convention on Climate Change (UNFCCC). Other anthropogenic sources (livestock, landfills, wastewater, rice, etc.) are from the EDGAR v4.3.2 inventory in 2012 as global default [Janssens-Maenhout et al., 2019] and from the gridded version of the US Environmental Protection Agency (EPA) greenhouse gas inventory in 2012 for the continental US [Maasakkers et al., 2016]. Seasonalities of rice and manure emissions are based on B. Zhang et al. [2016] and Maasakkers et al. [2016], respectively.


We use monthly wetland methane emissions in 2019 from the 18-member ensemble mean of the WetCHARTs version 1.3.1 inventory [Bloom et al., 2017], which has good performance in reproducing the observed wetland methane seasonal cycle for most regions [Parker et al., 2020b]. Other natural sources include open fire emissions in 2019 from the Global Fire Emissions Database version 4 (GFED4) [van der Werf et al., 2017], termite emissions from Fung et al. [1991], and geological seepage

from Etiope et al. [2019] scaled to the global magnitude of 2 Tg a⁻¹ from Hmiel et al. [2020]. The total methane sources in the prior estimate add up to 542 Tg a⁻¹, which is smaller than the bottom-up inventory estimate of 594-881 Tg a⁻¹ from the Global Methane Budget 2020 [Saunois et al., 2020]. The difference is mainly caused by the higher estimates of emissions from freshwater (117 – 212 Tg a⁻¹), seeps (18 – 65 Tg a⁻¹), oil and gas (72 – 97 Tg a⁻¹), and coal (29 – 61 Tg a⁻¹) in Saunois et al. [2020]. The freshwater source in our prior estimate is included in the wetland sector as represented by WetCHARTs [Bloom

et al., 2017].

The main sink of methane is oxidation by the hydroxyl radical (OH) in the troposphere [Ehhalt and Heidt, 1973], with a corresponding lifetime of $11.2 \pm 1.3$ years as constrained by the methylchloroform proxy [Prather et al., 2012]. Our prior estimate for the loss of methane from reaction with tropospheric OH is calculated using archived 3-D climatological monthly

fields of OH concentrations from a GEOS-Chem full-chemistry simulation [Wecht et al., 2014], yielding a methane lifetime of 10.5 years due to oxidation by tropospheric OH. Additional minor losses include oxidation by tropospheric Cl atoms computed using archived Cl concentration fields from Wang et al. [2019], stratospheric oxidation computed with archived 2-D monthly loss frequencies from the NASA Global Modeling Initiative model [Murray et al., 2012], and soil uptake of methane specified following Murguia-Flores et al. [2018].




## 3.2 Analytical inversion

We apply Bayesian inference to optimize a state vector consisting of (1) annual mean non-wetland methane emissions for land-containing 2°×2.5° grid cells (4020 state vector elements), (2) monthly wetland methane emissions for the 14 subcontinental regions of Figure 4 (168 elements), and (3) annual hemispheric tropospheric OH concentrations (2 elements).

This setup is the same as in Lu et al. [2021] and Y. Zhang et al. [2021] except for the higher horizontal resolution applied to non-wetland emissions. Together we have 4190 state vector elements, which requires a total of 4190 perturbed GEOS-Chem simulations and a base simulation to construct the full Jacobian matrix. This is readily done on a high-performance computing platform as an embarrassingly parallel workload. Initial conditions on January 1, 2019 are obtained from a standard GEOS-Chem simulation using the prior emission estimates and a 10-year spin-up, and are scaled by a globally uniform factor of 0.97

in order to match the global mean column mixing ratio retrieved from TROPOMI between Jan 1 and Jan 10, 2019. This initialization is used for both TROPOMI and GOSAT inversions.

The posterior estimate as defined by Bayesian optimization assuming Gaussian error statistics is obtained by minimizing the scalar cost function $J(x)$:


$$J(x) = (x - x_a)^T \mathbf{S_a}^{-1}(x - x_a) + \gamma (y - \mathbf{K}x)^T \mathbf{S_o}^{-1}(y - \mathbf{K}x), \tag{3}$$

where $\mathbf{K}$ is the Jacobian matrix describing the sensitivity of the observations to the state vector as simulated by GEOS-Chem, $\mathbf{S_a}$ is the prior error covariance matrix, $\mathbf{S_o}$ is the observational error covariance matrix assumed to be diagonal, and $\gamma$ is

a regularization parameter that accounts for the effect of unresolved correlation in the observational error.

$\mathbf{S_a}$ is constructed by assuming 50% prior error standard deviation for all non-wetland emissions on the 2°×2.5° grid and 10% prior error standard deviation for hemispheric annual mean OH concentrations, with no error correlations. Prior error variances and covariances for monthly wetland emissions in the 14 subcontinental regions are calculated using the WetCHARTs model

ensemble [Bloom et al., 2017] following Y. Zhang et al. [2021].

Observational error variances (diagonal elements of $\mathbf{S_o}$) are calculated using the residual error method [Heald et al., 2004] as the variance of the residual difference between observations and the GEOS-Chem prior simulation on the 2°×2.5° grid after subtracting the mean difference. This method sums up errors from instrument retrieval, representation, and GEOS-Chem

transport. We find a global annual mean error of 13 ppbv for TROPOMI and 14 ppbv for GOSAT. For cases where the calculated error is smaller than the instrument precision reported in the satellite retrieval, we use the latter instead (annual means of 9 ppbv for GOSAT and 2 ppbv for TROPOMI).





**S$_o$** is specified as diagonal but there is in fact some observational error covariance if only from the GEOS-Chem transport. For

TROPOMI in particular, there may be many individual observations for a single GEOS-Chem grid cell and time, and the corresponding transport errors would be perfectly correlated. Although one could average all TROPOMI observations within a 2°×2.5° grid cell before ingesting them in the inversion, this would lose the averaging kernel specificity for each observation. We therefore use a regularization parameter $\gamma$ [Hansen et al., 1999; Y. Zhang et al., 2018, 2020; Maasakkers et al., 2019; Lu et al., 2021] to account for the off-diagonal structure missing in **S$_o$**. Based on the corner of the L-curve [Hansen et al., 1999]

and the expected chi-square distribution of the cost function [Lu et al., 2021] (see Figure S2), we choose $\gamma = 0.002$ for TROPOMI observations and $\gamma = 0.5$ for the GOSAT observations. The regularization parameter of 0.5 for GOSAT is larger than the values of 0.05 - 0.1 in Maasakkers et al. [2019], Y. Zhang et al. [2021], and Lu et al. [2021], which used a 4°×5° resolution and several years of observations. The smaller value of $\gamma$ for TROPOMI is due to its large number of collocated observations on the 2°×2.5° model grid. Shen et al., [2021] conducted a regional inversion of TROPOMI data using GEOS-

Chem at 0.25°×0.3125° resolution and found that $\gamma = 0.25$ provided the best fit to the L-curve, reflecting the much smaller number of collocated observations on the 0.25°×0.3125° grid.

We further balance the prior terms in the cost function by weighing the wetland emission term by the number of elements in the state vector (4020/14). This step ensures that changes in non-wetland and wetland emissions are equally expensive from a

cost-function perspective [Maasakkers et al., 2019]. Similarly scaling the hemispheric OH terms in the cost function by the number of elements in the state vector (4020/2) would lead to excessively small posterior adjustments. We therefore choose weighting factors of the OH terms (400 for TROPOMI, 450 for GOSAT) that lead to a standard deviation of 5% in the posterior OH adjustments.

There is some arbitrariness in the selection of regularization parameters $\gamma$ and prior weighting factors in the inversion. We examined the sensitivity to the choice of $\gamma$ with sensitivity inversions using (1) $\gamma = 0.02$ and (2) $\gamma = 0.5$ for TROPOMI, and (1) $\gamma = 0.02$ and (2) $\gamma = 0.002$ for GOSAT. We further examined the sensitivity to the choice of weighting factors with sensitivity inversions using (3) no weighting factors, (4) a weighting factor of 1 for wetland terms, and (5) a weighting factor of 2010 for the OH terms (i.e., the ratio of the number of state vectors in non-wetland and OH terms). In this manner we

performed ensembles of 5 sensitivity inversions using TROPOMI observations only, 5 using GOSAT observations only, and 35 using the joint TROPOMI and GOSAT observations.

The best posterior estimate obtained by minimization of the cost function $J(\boldsymbol{x})$ is given by [Rodgers, 2000]:


$$\hat{\boldsymbol{x}} = \boldsymbol{x}_a + \left(\gamma \mathbf{K}^T \mathbf{S_o}^{-1}\mathbf{K} + \mathbf{S_a}^{-1}\right)^{-1}\gamma \mathbf{K}^T \mathbf{S_O}^{-1}(\boldsymbol{y} - \mathbf{K}\boldsymbol{x}_a). \tag{4}$$



with posterior error covariance matrix $\hat{\mathbf{S}}$:

$$\hat{\mathbf{S}} = (\gamma \mathbf{K}^T \mathbf{S_o}^{-1} \mathbf{K} + \mathbf{S_a}^{-1})^{-1}. \tag{5}$$


The averaging kernel matrix $\mathbf{A}$ defines the sensitivity of the solution to the true state:

$$\mathbf{A} = \mathbf{I} - \hat{\mathbf{S}} \mathbf{S_a}^{-1}, \tag{6}$$

where $\mathbf{I}$ is the identity matrix. The trace of $\mathbf{A}$ represents the number of independent pieces of information on the state vector that is gained from the observations, and is called the degrees of freedom for signal (DOFS) [Rodgers, 2000].

The posterior solution can also be presented in reduced dimensionality. For instance, posterior emissions on the 2°×2.5° grid can be aggregated to national or global emissions from individual source sectors. This aggregation can be expressed with a
summation matrix $\mathbf{W}$ to represent the linear transformation from the full state vector to the reduced state vector. The posterior estimate of the reduced state vector ($\hat{x}_{red}$) is computed as

$$\hat{x}_{red} = \mathbf{W}\hat{x}. \tag{7}$$

and its posterior error covariance and averaging kernel matrices are given by


$$\hat{\mathbf{S}}_{red} = \mathbf{W}\hat{\mathbf{S}}\mathbf{W}^{\mathbf{T}}, \tag{8}$$
$$\mathbf{A}_{red} = \mathbf{W}\mathbf{A}\mathbf{W}^*, \tag{9}$$

where $\mathbf{W}^* = \mathbf{W}^{\mathbf{T}}(\mathbf{W}\mathbf{W}^{\mathbf{T}})^{-1}$ is the Moore-Penrose inverse [Calisesi et al., 2005].

**4 Results and discussion**

Our discussion focuses principally on results from the base inversions of the TROPOMI-only, GOSAT-only, and joint data sets, and uses ranges from the inversion ensemble as a more conservative estimate of posterior errors than the posterior error covariance matrix $\hat{\mathbf{S}}$. In this analysis we exclude ensemble members with unreasonable emission adjustments (e.g., negative emissions aggregated at regional scales) and OH adjustments larger than 40% (see Table S1, S2).






## 4.1 Information content from the inversions

Figure 5 shows the corrections to the prior estimates of non-wetland emissions (posterior/prior ratios) on the 2°×2.5° grid for the TROPOMI, GOSAT, and TROPOMI+GOSAT inversions. These corrections will be discussed in Section 4.3. Also shown are the averaging kernel sensitivities of the inversions, defined as the diagonal elements of the averaging kernel matrices and
representing the ability of the observations to determine the posterior solution independently of the prior estimate (1 = fully, 0 = not at all). The averaging kernel sensitivities are highest over major anthropogenic source regions where the methane enhancements are the largest.

The TROPOMI inversion has 155 DOFS, meaning that it contains 155 independent pieces of information on the distribution
of methane emissions and OH concentrations. The GOSAT inversion has 238 DOFS, more than TROPOMI despite having much fewer observations. This reflects the large error correlation between individual TROPOMI observations on the 2°×2.5° grid of the inversion, as expressed by the difference between the regularization parameters for GOSAT observations ($\gamma = 0.5$) and TROPOMI observations ($\gamma = 0.002$). GOSAT with precise individual observations spaced by 250 km is particularly well adapted to an inversion on a 2°×2.5° grid. TROPOMI would be far more valuable in a regional inversion at higher spatial
resolution [Shen et al., 2021], although the regional biases discussed in Section 2 would still be a concern.

Zhang et al. [2021] previously reported an inversion of 2010-2018 GOSAT data using GEOS-Chem at 4°×5° resolution. That inversion achieved 179 DOFS, compared to 238 DOFS in our inversion for just one year of GOSAT data at 2°×2.5° resolution. The higher DOFS in our case reflects the higher dimension of our emission state vector (2°×2.5° versus 4°×5° grid cells),
combined with higher weight per observation ($\gamma = 0.5$ versus 0.05) because of lower error correlation on the 2°×2.5° scale. As pointed out above, the GOSAT data are particularly well suited to a 2°×2.5° resolution for the inversion. The finer 2°×2.5° resolution also allows for improved sectoral and national attribution of inversion results as will be done in Section 4.3.

In the joint inversion, TROPOMI observations add additional DOFS to the GOSAT posterior at 0°–30° N (mainly over India
and the Middle East), where TROPOMI has more observations than in the rest of the world (Figure 1). TROPOMI has lower averaging kernel sensitivities at 30°–60° N and 0°–60° S, and the information content over these two regions mostly comes from GOSAT. This could reflect the limitation of using a single global regularization parameter $\gamma$ for the TROPOMI observations, because the observations should have more weight (larger $\gamma$) when they are less dense. Improving this aspect of the inversion is a target for future work.


The 155 DOFS for TROPOMI are partitioned as 151 for non-wetland emissions, 3 for wetlands, and 1 for OH. The 238 DOFS for GOSAT are partitioned as 232 for non-wetland emissions, 5 for wetlands, and 1 for OH. The wetland emissions are largely unchanged in both inversions, because of error weighting in the cost function that penalizes departure from the prior estimate.



Without this error weighting, the TROPOMI inversion would yield unrealistic wetland emissions and seasonalities (case 3 in
Table S1). The problem may reflect systematic biases in the TROPOMI retrieval due to the low SWIR surface albedo over
wetland surfaces (e.g., Brazil and central Africa, see Figure S3, and boreal wetlands in Canada and Russia), combined with
seasonal imbalance in observations (cloudiness for tropical wetlands, sun angle and snow for boreal wetlands). Improvement
in TROPOMI retrievals over wetlands is needed. In the meantime, our further discussion of results in Section 4.3 will focus
on the non-wetland emissions.


The posterior/prior ratio of global OH concentrations is 0.96 for both the TROPOMI and GOSAT inversions and 0.91 for the
joint inversion. Methane lifetimes against oxidation by tropospheric OH range from 10.7 to 11.0 years in the ensemble of
TROPOMI inversions excluding case 3 (Table S1) and from 10.7 to 11.1 years in the GOSAT inversions (Table S2). These
corrections improve agreement with the observationally constrained methane lifetime of $11.2 \pm 1.3$ years [Prather et al., 2012].
The north/south interhemispheric OH ratio (NH/SH) is 1.03 in the prior estimate, 0.93 from the TROPOMI inversion, 1.15
from the GOSAT inversion, and 1.03 in the joint inversion, suggesting that the observations do not usefully constrain this ratio.
Patra et al. [2014] estimated a ratio of $0.97 \pm 0.12$ from methyl chloroform observations.

### 4.2 Cross-fit to TROPOMI and GOSAT observations

Figure 6 shows the ability of the inversions to improve the fit between GEOS-Chem and the 2019 satellite observations when
using posterior versus prior emissions and OH concentrations. This includes cross-evaluation of the TROPOMI inversion with
independent GOSAT observations and vice versa. The simulation using prior emissions started on January 1, 2019 in an
unbiased state compared to TROPOMI and a -1.7 ppbv global bias relative to GOSAT (Section 2). It underestimates 2019
GOSAT observations everywhere by an average of 14.6 ppbv (Figure 6), implying the need to increase methane sources and/or
decrease OH concentrations. It also underestimates TROPOMI over most of the world but overestimates in some regions
(notably the subarctic) that may reflect TROPOMI retrieval biases as discussed in Section 2.

Both TROPOMI and GOSAT inversions reduce the negative differences between simulations and observations. The
improvement can be measured by the value of the cost function $J(x)$ in Equation 3, which decreases by 35% for the TROPOMI
inversion and 54% for the GOSAT inversion. Cross-evaluation of the posterior simulation with the independent data set
(TROPOMI or GOSAT) also shows improvement. The fit to the GOSAT data is improved everywhere even with the
TROPOMI inversion. TROPOMI shows problematic regions where the inversion overcorrects the prior bias. This will be
discussed further in Section 4.3.



### 4.3 Implications for methane emissions

#### 4.3.1 Global distribution

Our posterior/prior ratios for the 2019 GOSAT inversion in Figure 5 show large upward adjustments of non-wetland emissions in the south-central US, Venezuela, and the Middle East, consistent in magnitude with the previous inversion of 2010-2018 GOSAT data by Zhang et al. [2021], who used the same prior estimate. These two inversions also have consistent magnitude of downward adjustments in the western US, Europe, Russia, and North China Plain. We find larger upward adjustments than Zhang et al. [2021] in India, East Africa, and Brazil, which they identified as regions with rapidly increasing emissions over

the 2010-2018 period.

Figure 5 shows agreement between GOSAT and TROPOMI in the adjustments of methane emissions in several major source regions including western Russia, the North China Plain, the south-central US, East Africa, and Venezuela. A few regions have adjustments of different signs, notably Brazil and parts of central Africa where the TROPOMI retrievals are likely biased

(Figure 3 and Figure S3).

We conducted a global sectoral breakdown of the posterior non-wetland emission fluxes on the $2^{o} \times 2.5^{o}$ grid by using Equation 7, where we assume that the partitioning between sectors in a given grid cell to be correct in the prior inventory and that the posterior/prior ratio applies equally to all sectors in the grid cell. This follows the same methodology previously used by

Maasakkers et al. [2019] and Zhang et al. [2021] for GOSAT, but our use of the finer $2^{o} \times 2.5^{o}$ grid makes sectoral attribution more accurate. On the other hand, our restricted adjustment of wetland emissions means that errors in wetland emissions could be projected to non-wetland sectors.

Table 1 compiles these sectoral attributions of inversion results. Of particular interest is the oil/gas sector, for which the global

prior estimate (66 Tg a$^{-1}$) is based on 2016 UNFCCC national inventory reports. We find global decreases in the joint TROPOMI+GOSAT inversion to 56 Tg a$^{-1}$, largely driven by decreases in Russia. This is consistent with the correction in Russian oil/gas emissions reported to the UNFCCC, from 27 Tg a$^{-1}$ in the communication used by the GFEI to 16 Tg a$^{-1}$ in the latest communication [UNFCCC, 2020]. Livestock emissions (the single largest anthropogenic methane source) are adjusted upward by the joint inversion from 116 Tg a$^{-1}$ in the EDGAR v4.3.2 prior estimate to 139 Tg a$^{-1}$.

#### 4.3.2 Major source regions

Figure 7 shows emissions in the top five anthropogenic methane source regions including China, India, Brazil, Europe, and the contiguous US (CONUS). These regions account for 56% of global posterior anthropogenic emissions in the GOSAT inversion.



In China, both GOSAT and TROPOMI inversions adjust non-wetland methane emissions downward in the North China Plain. This has been a long-standing result of inversions of satellite data using EDGAR as prior estimate [Monteil et al., 2013; Thompson et al., 2015; Alexe et al., 2015; Turner et al., 2015] and has been attributed to an overestimate of emissions from the coal sector which dominates total EDGAR emissions in the region. More recent inversions using the UNFCCC-based GFEI as prior estimate have found the same result [Lu et al., 2021; Y. Zhang et al., 2021], but GFEI takes its spatial allocation of

coal emissions from EDGAR.  A more detailed bottom-up analysis of Chinese coal emissions by Sheng et al. [2019] finds most of the emissions to be in South China, in contrast to EDGAR which places them in the North China Plain. Our TROPOMI inversion over Southeast China shows spatially inconsistent results (Figure 5) and overcorrects the fit to observations (Figure 6), which may be due to aliasing between coal and rice emissions. Rice cultivation is the dominant source of methane in Southeast China in our prior estimate, but the emissions have large seasonality and peak in summer when cloudiness is

pervasive and TROPOMI observations are few, as shown in Figure 8. GOSAT is less affected by cloudiness (Figure 8), on account of its use of the $CO_2$ proxy retrieval method. We therefore exclude posterior estimates from TROPOMI and the joint inversions from Figure 7. Because China accounts for a large fraction of global rice [Chen et al., 2013] and coal emissions [Cheng et al., 2011; Miller et al., 2019], we also exclude these entries from Table 1. At national scale, the GOSAT inversion adjusts anthropogenic methane emissions downward from 67 Tg a$^{-1}$ to 56 Tg a$^{-1}$ in China, very close to the value of 55 Tg a$^{-1}$

reported by China to the UNFCCC in 2014 [UNFCCC].

All three inversions adjust methane emissions upwards in India. The results from the base inversion are at the higher end of the range from the inversion ensemble, but the 41-55 Tg a$^{-1}$ range of national emissions spanned by the ensemble is still much higher than previous inversions of GOSAT and in-situ data including 33 Tg a$^{-1}$ for 2010-2018 by Y. Zhang et al. [2021] and

22 Tg a$^{-1}$ for 2010-2015 by Ganesan et al. [2017]. This may reflect the rapid increase of Indian emissions over the 2010-2018 period previously identified by Y. Zhang et al. [2021] and attributed principally to livestock.

In Brazil, the large upward adjustments from 23 Tg a$^{-1}$ (prior) to 37 Tg a$^{-1}$ (GOSAT) and 33 Tg a$^{-1}$ (joint) in the posterior estimates are consistent with previous top-down estimates [Maasakkers et al., 2019; Y. Zhang et al., 2021]. TROPOMI shows

adjustments in the opposite direction, likely reflecting observational bias associated with low SWIR surface albedo (Figure 3) and limited number of observations. The joint inversion is dominated by results from GOSAT on account of the much higher averaging kernel sensitivities for the inversion (Figure 5).

All inversions adjust emissions downwards in Europe (prior: 39 Tg a$^{-1}$, TROPOMI: 27 Tg a$^{-1}$, GOSAT: 34 Tg a$^{-1}$, joint: 21 Tg

a$^{-1}$), consistent with the previous downward adjustments in the 2010-2018 mean in the GOSAT inversion and the small negative trend in methane emissions [Y. Zhang et al., 2021]. The largest reductions of emissions in Europe are from coal and oil/gas emissions.



In CONUS, the large upward adjustment in the south-central region reflects the well-known underestimate of oil/gas emissions
by the US EPA inventory in that region [Kort et al., 2014; Smith et al., 2017; Peischl et al., 2018; Alvarez et al., 2018;
Maasakkers et al., 2021; Gorchov Negron et al., 2020; Y. Zhang et al., 2021; Lyon et al., 2021]. The posterior estimates from
both TROPOMI and GOSAT adjust national methane emissions slightly downwards from 31 Tg a$^{-1}$ to 27 Tg a$^{-1}$ (TROPOMI)
and 30 Tg a$^{-1}$ (GOSAT) over CONUS, close to the posterior estimates of 31 Tg a$^{-1}$ from the 0.5°×0.625° inversion over 2010-
2015 [Maasakkers et al., 2021]. The joint inversion adjusts emissions slightly upwards to 34 Tg a$^{-1}$ due to the larger sensitivity
over the south-central US.

## 5 Conclusions

We used one year (2019) of atmospheric methane column observations from the new TROPOMI satellite instrument in a
global inverse analysis of methane sources at 2°×2.5° resolution, and compared results to the same analysis using the more
mature but sparser GOSAT instrument as well as the combination of the two instruments. By analytical solution to the inverse
problem, we were able to quantitatively compare the information content from the two satellite datasets as top-down constraints
on methane emissions. This includes averaging kernel sensitivities and degrees of freedom for signal (DOFS) that quantify the
number of independent pieces of information on the distribution of methane emissions.

We began by validating the global observations from TROPOMI and GOSAT by common reference to the ground-based
TCCON methane column measurements, using the GEOS-Chem CTM to correct for the effects of different prior estimates
and averaging kernels in the retrievals from each instrument. Results show that TROPOMI and GOSAT are globally biased
by -2.7 ppbv and -1.0 ppbv respectively. Their regional biases relative to TCCON are 7 ppbv and 3 ppbv, respectively,
sufficiently small for inverse analyses of methane emissions on regional to global scales. Intercomparison between TROPOMI
and GOSAT shows larger regional differences exceeding 20 ppbv, generally in places where the SWIR surface albedo is low
and TROPOMI retrievals would be subject to biases [Lorente et al., 2021]. GOSAT is less sensitive to albedo-driven biases
because of its $CO_2$ proxy retrieval method, as compared to the full-physics retrieval in TROPOMI.

We find that the GOSAT inversion has a global DOFS of 232 for non-wetland methane emissions on the 2×2.5° grid, larger
than the TROPOMI inversion (DOFS of 151) despite the TROPOMI data being much denser. This is because individual
TROPOMI observations have large error correlations on the 2°×2.5° grid of the inversion, whereas the GOSAT observations
with their 250 km separation are ideally suited for that scale. Finer-scale inversions, as done for regional studies, would be far
more effective at exploiting the information from TROPOMI. A better representation of error correlation, accounting for the
relative sparsity of TROPOMI data in cloudy regions, would also increase the value of TROPOMI data in global inversions.
Combining the TROPOMI and GOSAT data in a joint inversion increases the DOFS to 244, with most of the added information
from TROPOMI in the 0°–30° N latitudinal band including India and the Middle East.



The TROPOMI and GOSAT inversions for 2019 show consistent upward adjustments of anthropogenic methane emissions over Venezuela (oil/gas) and the south-central US (oil/gas), and downward adjustments over Europe (oil/gas and coal), Russia (oil/gas), and North China Plain (coal). These adjustments are relative to the official national inventory reports to the UNFCCC in 2016 and used as prior estimates in our inversion. The TROPOMI and GOSAT inversions also show consistent upward adjustments over East Africa where livestock emissions are large. Global livestock emissions increase from 116 Tg a$^{-1}$ in the EDGAR v4.3.2 prior estimate to 139 Tg a$^{-1}$ in the joint GOSAT+TROPOMI inversion. Some regions show large inconsistencies between TROPOMI and GOSAT inversions, and we find that these generally reflect TROPOMI regional biases in low-albedo regions. The strict cloudiness filter used in TROPOMI observations is also problematic in methane source regions such as wetlands and rice agriculture that have extensive and sometimes seasonal cloud cover.

Our results demonstrate the potential of applying TROPOMI observations to constrain methane emissions on a global scale through inverse analyses, but also stress the need for caution. The methane retrieval from TROPOMI is still in an early stage, and the current operational product appears to have systematic biases in low-albedo regions. Future generations of the retrieval may address these data quality flaws [Lorente et al., 2021]. Improved accounting of model transport error correlations in global coarse-resolution inversions is also needed to fully exploit the TROPOMI observations on a global scale. In the meantime, GOSAT provides a high-quality record of methane observations going back to 2010 and we have shown that 1 year of GOSAT observations can usefully inform emissions on a 2°×2.5° grid. GOSAT will thus be increasingly useful in the future to attribute methane trends and to validate future generations of the TROPOMI retrieval.

**Acknowledgments**

This work was funded by the NASA Carbon Monitoring System under NASA award number 80NSSC18K0178 to Harvard University. RJP is funded via the UK National Centre for Earth Observation (NE/N018079/1). We thank the Japanese Aerospace Exploration Agency, National Institute for Environmental Studies, and the Ministry of Environment for the GOSAT data and their continuous support as part of the Joint Research Agreement. This research used the ALICE High Performance Computing Facility at the University of Leicester for the GOSAT retrievals.

**Data availability statement**

The TROPOMI methane observations are downloaded from http://www.tropomi.eu/data-products/methane (last accessed Aug 8, 2020). The GOSAT methane retrieval is available at https://catalogue.ceda.ac.uk/uuid/18ef8247f52a4cb6a14013f8235cc1eb



(last accessed Dec 29, 2020). The TCCON measurements are downloaded from https://data.caltech.edu/records/293 (last access Sep 6, 2020).

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



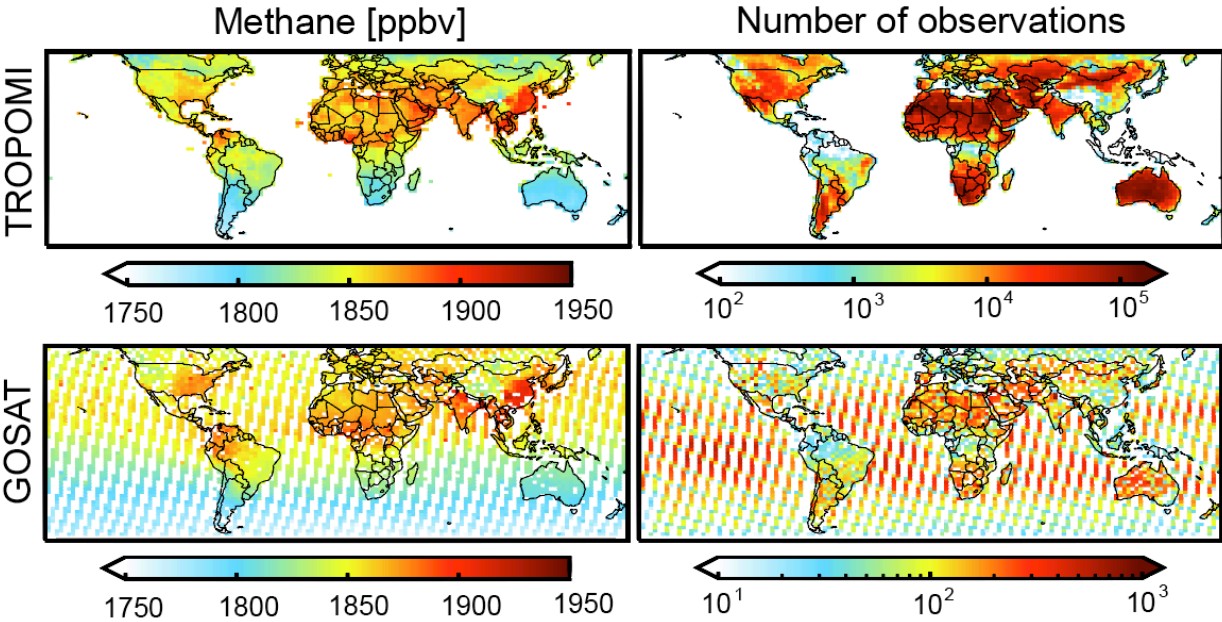


**Figure 1.** Mean column averaged dry methane mixing ratios ($X_{CH4}$) measured by TROPOMI and GOSAT in 2019, and number of observations from each instrument in that year. The data have been filtered using "qa_value" ≥ 0.5 for TROPOMI and "xch4_quality_flag" = 0 for GOSAT, and are shown on the GEOS-Chem 2º×2.5º grid. Note the difference in scale for the number of observations by TROPOMI and GOSAT.


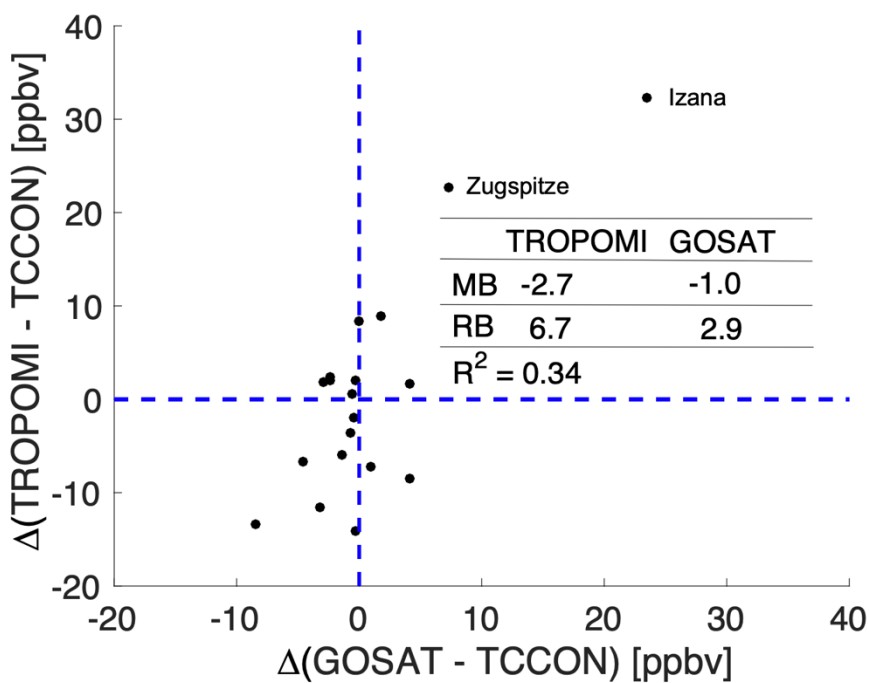

Figure 2. Biases of TROPOMI and GOSAT methane ($X_{CH4}$) retrievals relative to TCCON. Values are averages for May 2018 - April 2019 at each of the 22 sites of the TCCON network, and have been corrected for differences in averaging kernels and prior vertical profiles on the 2°×2.5° GEOS-Chem grid as described in the text. The large correlated biases at Zugspitze and Izana can be explained by the high altitude of these TCCON sites. Statistics for the other 20 sites are given inset including the mean bias (MB), the regional bias (RB) calculated as the standard deviation of the bias between satellite and individual TCCON stations, and the coefficient of determination ($R^2$) between the TROPOMI and GOSAT biases.

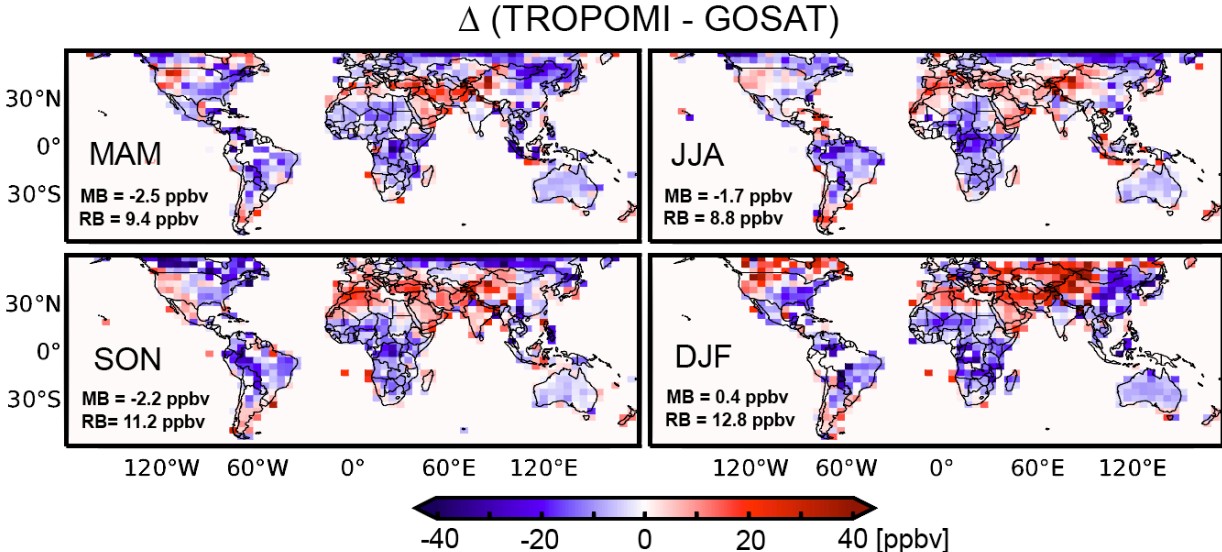

**Figure 3. Seasonally averaged differences (Δ) of $X_{CH4}$ between TROPOMI and GOSAT retrievals for May 2018 – April 2019 on a 4°×5° grid. The retrievals have been corrected for differences in averaging kernels and prior vertical profiles as described in the text. MB is the global mean bias of TROPOMI relative to GOSAT and RB is the regional bias as defined by the standard deviation of Δ on the 4°×5° grid.**





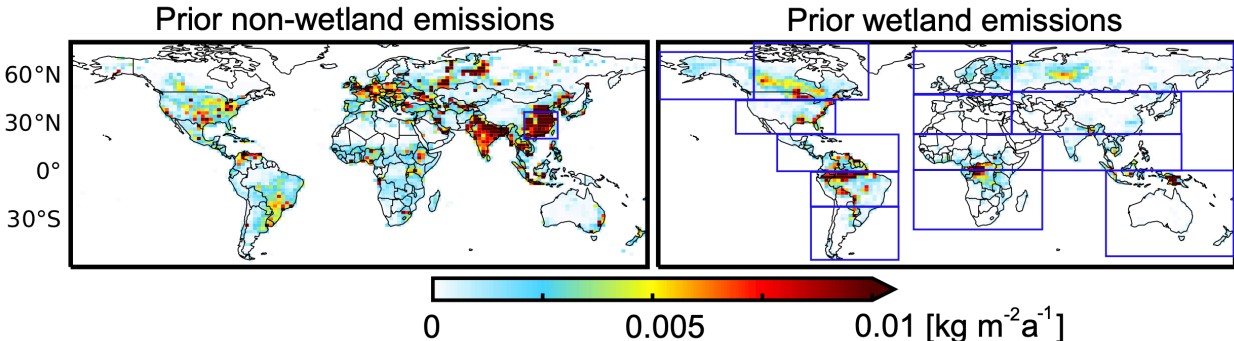


**Figure 4. Spatial distribution of prior methane emissions in 2019. The blue box over China in the left panel indicates the region used for seasonality analysis in Figure 8. The blue boxes in the right panel indicate the 14 subcontinental regions of Y. Zhang et al. [2021] for which monthly wetland emissions are aggregated to limit the dimensions of the inversion state vector.**


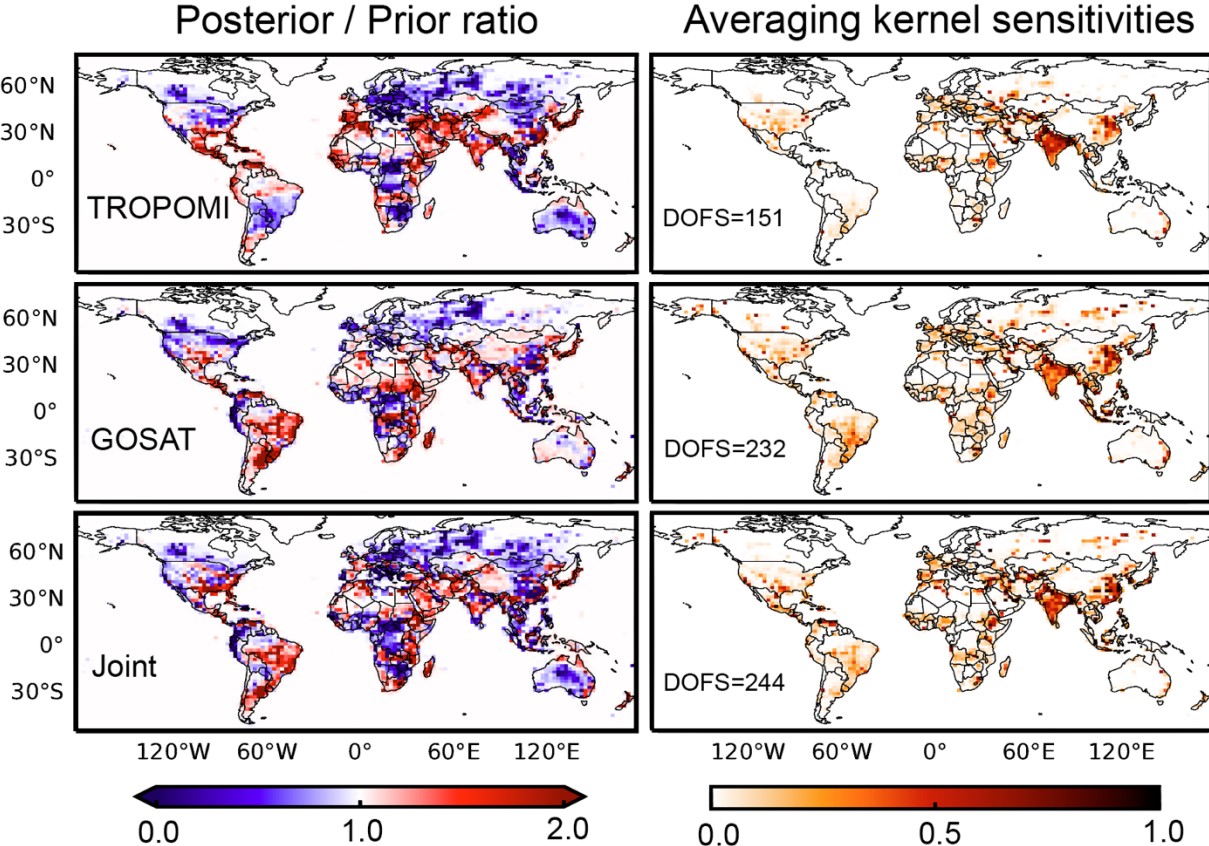

**Figure 5. Corrections to prior estimates of 2019 non-wetland methane emissions on the 2º×2.5º grid (posterior/prior ratios), and corresponding averaging kernel sensitivities. Results are shown for the TROPOMI, GOSAT, and joint TROPOMI+GOSAT inversions. The averaging kernel sensitivities are the diagonal elements of the averaging kernel matrix for the inversion, and measure the ability of the observations to constrain the emissions (1 = fully, 0 = not at all). The sum of averaging kernel sensitivities (trace of the averaging kernel matrix) defines the degrees of freedom for signal (DOFS) for the inversion, shown inset. DOFS including contributions from wetland emissions and OH concentrations are 155 for TROPOMI and 238 for GOSAT.**






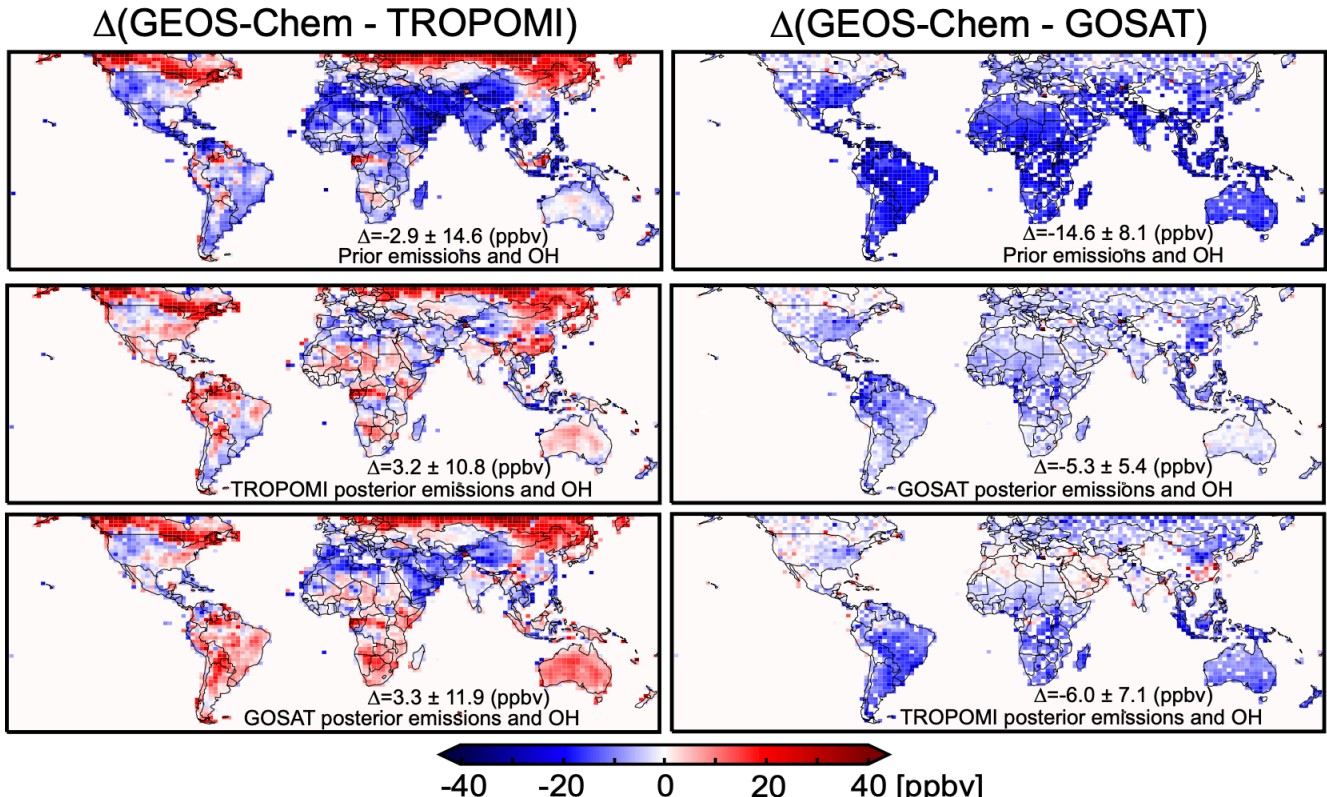

**Figure 6. Comparison of GEOS-Chem $X_{CH4}$ to TROPOMI and GOSAT observations. Panels show annual mean differences for 2019 between the GEOS-Chem simulation and observations, with mean bias ± standard deviation given inset. Top panels: GEOS-Chem with prior emission and OH estimates. Middle panels: GEOS-Chem with posterior estimates from the TROPOMI inversion. Bottom panels: GEOS-Chem with posterior estimates from the GOSAT inversion.**

**Figure 7. Annual anthropogenic methane emissions in 2019 for five major source regions, accounting for 56% of global anthropogenic emissions in the inversion of GOSAT data. The vertical bars represent the range of posterior emissions from the ensemble of inversions. Europe is defined as west of 37ºE. CONUS is the contiguous United States. TROPOMI and joint inversion results are not shown for China because of concern over biases resulting from seasonal cloudiness and prior errors in the spatial distribution of coal emissions (see text).**

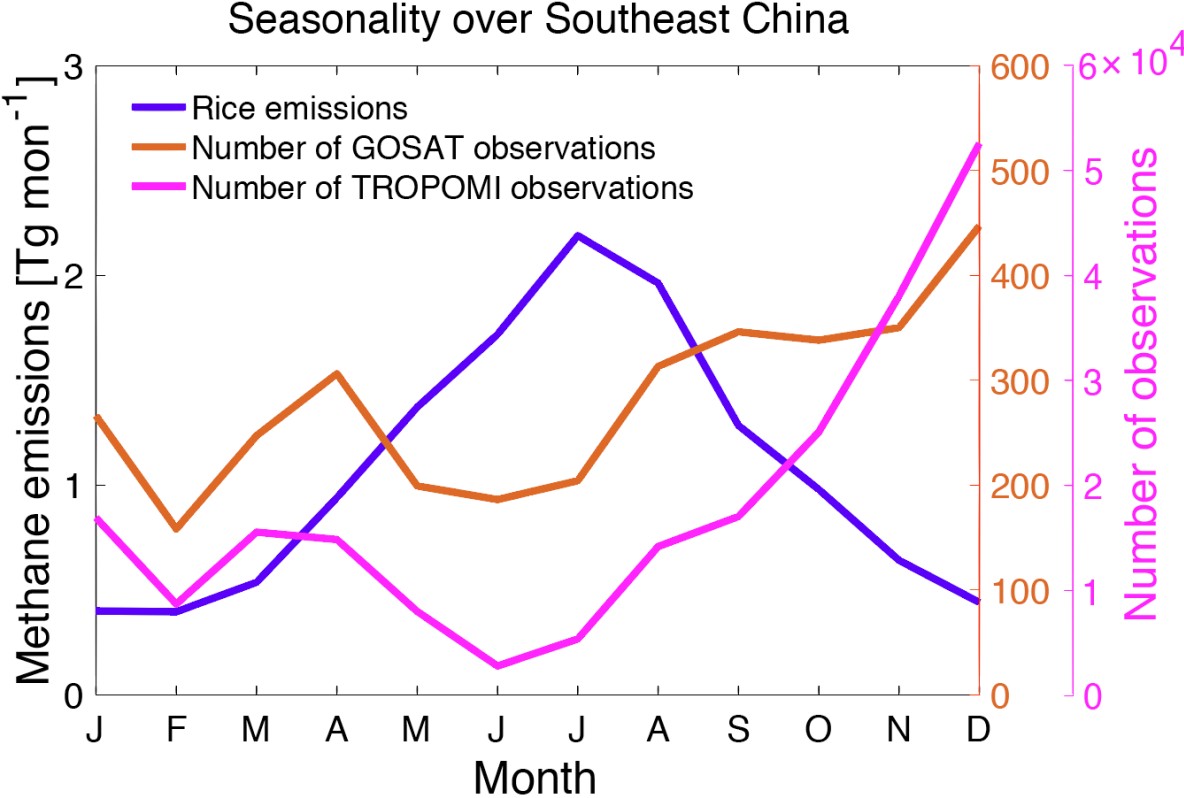

**Figure 8. Seasonality of methane emissions from rice cultivation and of the number of satellite observations over Southeast China (20°N – 37°N, 103°E – 123°E, shown in Figure 4) in 2019. The number of TROPOMI observations increases after August 2019 due to change in pixel size from 7 × 7 km² to 5.5 × 7 km². Seasonality of rice emissions is from B. Zhang et al. [2016]. Note the difference in scale for the number of observations by TROPOMI and GOSAT.**







**Table 1. Global methane budget for 2019.**

| | Prior estimate[a] [Tg a⁻¹] | Posterior estimates [Tg a⁻¹] | | |
| --- | --- | --- | --- | --- |
| | | TROPOMI | GOSAT | Joint |
| **Total sources** | **542** | **556** | **562** | **570** |
| Anthropogenic | 341 | 336 | 371 | 363 |
| Livestock | 116 | 126 | 143 | 139 |
| Oil and gas | 66 | 53 | 54 | 56 |
| Rice | 38 | NR [b] | 43 | NR [b] |
| Wastewater | 37 | 44 | 48 | 44 |
| Coal | 31 | NR [b] | 26 | NR [b] |
| Landfills | 30 | 27 | 30 | 31 |
| Other anthropogenic | 23 | 26 | 27 | 26 |
| Natural | 201 | 220 | 191 | 207 |
| Wetlands | 168 | 195 | 163 | 183 |
| Termites | 12 | 12 | 13 | 12 |
| Open fires | 19 | 11 | 13 | 10 |
| Seeps | 2 | 2 | 2 | 2 |
| **Total sinks** | **563** | **543** | **543** | **533** |
| Tropospheric OH | 489 | 468 | 468 | 458 |
| Soil uptake [c] | 34 | 34 | 34 | 34 |
| Stratospheric loss [c] | 35 | 35 | 35 | 35 |
| Tropospheric Cl [c] | 6 | 6 | 6 | 6 |
| **Imbalance** | **-21** | **13** | **19** | **37** |
| **Lifetime against tropospheric OH [a][d]** | 10.5 | 11.1 | 11.1 | 11.3 |

[a] Prior anthropogenic source estimates are from EDGAR v4.3.2 [Janssens-Maenhout et al., 2017] in 2012, superseded by oil, gas, and coal emissions from GFEI [Scarpelli et al., 2020] for 2016, and gridded EPA inventory data for the US [Maasakkers
et al., 2016]. Prior wetland emissions in 2019 are from WetCHARTs [Bloom et al., 2017]. Open fire emissions are from the Global Fire Emissions Database version 4 (GFED4) in 2019 [van der Werf et al., 2017]. Termite emissions are from Fung et al. [1991]. Geological seepages are from Etiope et al. [2019] scaled to the global magnitude from Hmiel et al. [2020].
[b] Not reported because of the potential for bias in the sectoral attribution of TROPOMI inversion results for China, which is a major global source of emissions from rice and coal. See text for details.
[c] These minor sinks are not optimized by the inversion.
[d] Lifetime of total atmospheric methane against oxidation by tropospheric OH