# Peer review of "Global distribution of methane emissions: a comparative inverse analysis of observations from the TROPOMI and GOSAT satellite instruments"

_Atmospheric Chemistry and Physics, 2021_

## Author Comment (AC1)

Reply on RC1

Thanks for the feedback on this work. We have responded to each reviewer comment below. Our replies are in blue, and the revised manuscript text is written in bold.

This work estimates the global methane emission for the year 2019 at 2°x2.5° spatial resolution based on inverse modeling technique by utilizing the TROPOMI and GOSAT satellite observations. It validates TROPOMI and GOSAT observation against TCCON methane column measurements, using the GEOS-Chem CTM and also provides the sensitivities parameters such as averaging kernel and degrees of freedom for signal which quantify the number of independent pieces of information on the distribution of methane emissions. The paper provides the discussion on estimated methane emission at different major methane source regions around the world using GOSAT, TROPOMI, and joint data-set. I recommend its acceptance after the minor revision with following specific/minor comments:

specific comments:

1). There are techniques such a 4-dimentional variational data assimilation (4D-var) and local ensemble Kalman filter (LETKF) that also provides grid-based flux estimation of methane by assimilating satellite observations. How close the inversion technique used in this study is to those techniques? I suggest author to add a paragraph that discuss the limitations of the high-resolution Bayesian inversion technique using satellite observations.

All these three methods are based on the Bayes theorem to optimize emissions given the methane observations and bottom-up emissions, as described in the 3rd paragraph of Section 1. 4D-Var approximates the inverse Hessian of the model and does not provide error statistics. LETKF updates emissions sequentially, but approximates the evolving of the covariance matrix and needs to assume a localization distance. The analytical Bayesian inversion used here does not have these approximations, but its limitation is the expensive computational cost to construct the whole Jacobian matrix and the requirement that the forward model is linear. We made the following modification to clarify:

"Most inverse analyses use **4-dimensional variational data assimilation (4D-Var) to solve** the Bayesian problem **numerically**, which enables inference of emissions at any resolution but does not readily provide error statistics [Meirink et al., 2008; Monteil et al., 2013; Wecht et al., 2014; Stanevich et al., 2019]. Analytical solution **is possible if the CTM is linear, as is the case for methane, and** has the advantage of including posterior error statistics and hence information content as part of the solution [Brasseur and Jacob, 2017]. It requires explicit construction of the Jacobian matrix of the CTM, **which is computationally expensive**, but this is readily done with massively parallel computing. Once the Jacobian matrix has been constructed, it can be applied to conduct ensembles of inversions at no added cost exploring the dependence of the solution on inversion parameters or observational data selection. **The analytical method can be applied as a Kalman Filter by updating methane emissions sequentially [e.g., Chen and Prinn, 2006; Fraser et al., 2013; Henne et al., 2016] but optimizing all emissions together over the period of interest makes the best use of the information content from the observations [Maasakkers et al., 2019; Lu et al, 2021; Y. Zhang et al., 2021]."**

2). The methane emission has been estimated using annual mean methane concentration data of GOSAT and TROPOMI. How would the seasonal variability of methane concentration affect such emission estimate? Is it possible to extend such inverse modeling set-up to estimate the methane emission at monthly scale?

We use all observations in a year to optimize methane emissions, but this is different from using the annual mean of methane concentrations since each observation is weighted by its error and provides its unique constraint on the emissions. We estimate wetland methane emissions at a monthly scale given its strong seasonal variations, as described in the first paragraph of Section 3.2. We added the following sentence to that paragraph to explain our motivation to optimize annual mean non-wetland emissions:

**"Trade-off is needed between spatial and temporal resolution in the state vector to avoid smoothing error in the inversion [Wecht et al, 2014] and for computational tractability. For non-wetland emissions we use high spatial resolution but only optimize the annual mean values because seasonality is relatively small and predictable. For wetland emissions, we cannot assume that the prior seasonality is correct [Maasakkers et al., 2019] and instead optimize monthly emissions at coarse spatial resolution."**

Minor comments:

Line 367, We conducted ……. grid cell.

How significant it is to apply equal ratios to all sectors in the grid cell. How better this is in comparison to isotopic fractionation method.

We added the following sentence after the cited one:

**"**We conducted a global sectoral breakdown … **This assumption is due to the lack of additional information (e.g., isotopic fractionation [Ghosh et al., 2015; Zhang et al., 2016; Zazzeri et al., 2017]) to separate different sources."**

Line 371, Study claimed that 2ox2.5o grid makes sectoral attribution more accurate, but we don't know true sectorial contribution.

We removed this sentence.

Line 385, In China,……………..Plain. Please cite the figure number here.

We changed the sentence to:

"In China, both GOSAT and TROPOMI inversions adjust non-wetland methane emissions downward in the North China Plain **(Figure 5)**."

Line 399, the analysis is performed for 2019, why did the author cite 2014 report.

The last reported year for China to the UNFCCC is 2014. We changed the sentence to:

"At national scale, … very close to the value of 55 Tg a$^{-1}$ **in the latest** report by China to the UNFCCC in 2014."

In Figure 3, large difference between GOSAT and TROPOMI can be seen during DJF at northward of 30oN. How does it affect inversion estimate?

We added the following sentence to the last paragraph of Section 2:

"The regional biases tend to be consistent across seasons, except for positive biases north of **40ºN** in DJF that could be associated with snow cover. **These biases may affect TROPOMI's constraints on the seasonal variations of methane sources.**"

We also made the following modifications in the 5$^{th}$ paragraph of Section 4.1:

"… the TROPOMI inversion would yield unrealistic wetland emissions and seasonalities (case 3 in Table S1). The problem may reflect systematic biases in the TROPOMI retrieval due to the low SWIR surface albedo over wetland surfaces (e.g., Brazil and central Africa, see Figure S4, and boreal wetlands in Canada and Russia), combined with seasonal imbalance in observations (cloudiness for tropical wetlands, sun angle and snow for boreal wetlands) **and seasonal biases at high northern latitudes (Figure 3).**"

Figure 7, Over India the estimate looks pretty close from all the methods, over Brazil, the joint inversion estimate is close to GOSAT, almost same story for Europe, but over CONUS the joint inversion is very high compared to both the inversion. How would you explain the joint inversion behavior over CONUS?

We made the following modifications in the last paragraph of Section 4.3.2:

"The joint inversion adjusts emissions upwards to **40** Tg a$^{-1}$ due to the larger **averaging kernel** sensitivity over the south-central US, **where emissions have large upward adjustments**."

---

## Author Comment (AC2)

Reply on EC1

Thanks for the feedback on this work. We have responded to each reviewer comment below. Our replies are in blue, and the revised manuscript text is written in bold.

Hi,

Nice work, if I understood correctly a set of TCCON sites was used for this work. Is it possible to provided which ones ?

Regards

We changed the sentence describing TCCON sites used in this work to:

"Only 9 TCCON sites have continuous observations for the whole year of 2019, but 2**1** sites **(Bialystok, Bremen, Burgos, California Institute of Technology, Darwin, Edwards, Garmisch, Izana, Jet Propulsion Laboratory, Sega, Karlsruhe, Lauder, Lamont, Orleans, Park Falls, Paris, Rikubetsu, Sodankyla, Tsukuba, Wollongong, and Zugspitze)** have observations over the period of May 2018 – Apr 2019 when TROPOMI observations started to be available."

---

## Author Comment (AC3)

Reply on RC2

Thanks for the feedback on this work. We have responded to each reviewer comment below. Our replies are in blue, and the revised manuscript text is written in bold.

The authors performed inversion analyses of atmospheric $CH_4$ using column average data from TROPOMI and GOSAT. Since the TROPOMI product is at the initial stage, the retrieved $CH_4$ data may have substantial uncertainties. Considering those uncertainties, this study elucidated characteristics of the TROPOMI data specifically for inverse results by comparing with the GOSAT data, which are at a mature stage. The inversions and related statistical calculations they performed revealed significant discrepancies between TROPOMI and GOSAT in terms of error characteristics and information content, and much better quality of the GOSAT data at this moment. Detail information of the TROPOMI $CH_4$ data obtained in this study is surely useful for improving the data quality of the TROPOMI data in a near future. The manuscript is well structured and most descriptions are clear to me. I think this paper can be published for Atmospheric Chemistry and Physics after minor revisions suggested below.

L43: "Climate change action" is better than "Climate action", isn't it?

Modified as suggested.

L45: Here, "GOSAT" first appears in the main text. Therefore, it should be written as "Greenhouse Gases Observing Satellite (GOSAT)"

Modified as suggested.

L53-54: Like "TROPOMI (GOSAT) observes light intensity at 2305-2385 (1630-1700) nm wavelength"

Modified as suggested.

L64: "inversion of a chemical transport model" => "inversion with a chemical transport model"

Modified as suggested.

L89-90: "one year of data" => "one year data"

Modified as suggested.

L105: TCCON first appears here. It should be "Total Carbon Column Observing Network (TCCON)". Furthermore, it is better to make a brief explanation of TCCON so that people who are not familiar in this field can understand the purpose of the use of TCCON.

We made the following modifications:

"We conducted … with ground-based **Total Carbon Column Observing Network (TCCON)** measurements ... **TCCON is a network of ground-based, sun-viewing, near Infra-Red, Fourier transform spectrometers to measure greenhouse gases [Wunch et al., 2011] and evaluate satellite retrievals [Parker et al., 2011; Butz et al., 2011; Houweling et al., 2014].**"

L141: Why was the 4x5 degree used only here (Fig. 3)? The other analyses seem to be done with the 2x2.5 degree.

We started the project by evaluating the two observations at 4°x5° resolution. We do not expect the bias to change when reducing the resolution to 2°x2.5°, so we did not redo this evaluation.

L201-202: I think this kind of pulse calculations assumes that the model is linear. Does the GEOS-Chem satisfies the model linearity?

We added the following sentence in the first paragraph of Section 3 to clarify:

"We use the GEOS-Chem global CTM … in the inversion. **The model is essentially linear except for a small nonlinearity from the optimization of OH concentrations [Maasakkers et al., 2019].**"

We also made the following modifications in the third paragraph of Section 1:

"Analytical solution **is possible if the CTM is linear, as is the case for methane, and** has the advantage of …"

L203: Only one-year-long inversion fluxes would have some errors attributed from the initial mole fraction field, which was optimized just by the globally uniform factor, especially for the earlier period. I think some discussion about that error is needed.

We added the changes in NMSE before and after correcting the initial condition in the sentence following the cited one:

**"**Initial conditions on January 1, 2019 are obtained from … This initialization **efficiently reduces the normalized mean square error (NMSE) between GEOS-Chem and TROPOMI observations on January 1, 2019 from 0.37 to 0.02 and** is used for both TROPOMI and GOSAT inversions**."**

L256: How is the "35" derived?

We changed the sentence to:

"In this manner we performed **6** inversions using TROPOMI observations only **(base inversion + 5 sensitivity inversions)**, **6 inversions** using GOSAT observations only, and **6 × 6 = 36 inversions** using the joint TROPOMI and GOSAT observations."

L266: The term "averaging kernel matrix" is familiar in the satellite retrieval field, but not the case for flux inversions. Although, it is theoretically correct, it might be better to put some note to avoid confusion.

We added the following sentence after Equation 6 to clarify:

**"Note that A here is different from the retrieval averaging kernel vectors in Section 2, which described the sensitivity of methane satellite retrievals to the vertical distribution of methane."**

L292: The ratios of the posterior/prior non-wetland emissions in Fig. 5 show values close-to-zero or over 2 in many places (e.g., over Europe and Africa in the joint case). Does this mean that the inversion zeroed/doubled non-wetland emissions? If that is the case, are the resulted emissions reasonable?

Doubled emission is not a problem since it is just $2\sigma$ ($\sigma = 50\%$) away from the prior. We added the following sentence to the title of Figure 5 to address the negative ratios:

"**Less than 3% grid cells have negative posterior / prior ratios, which is allowed by the statistics but is likely unphysical.**"

L314: It is hard to see the effect of TROPOMI. It would be better to show sensitivity differences from those of the GOSAT inversion case in the right bottom panel of Fig. 5.

We add a Figure in the Supporting Information to show the difference:

[Figure]

**Figure S3.** Differences between averaging kernel from the joint inversion and GOSAT inversion.

We change the description of the averaging kernel from the joint inversion to:

"In the joint inversion, TROPOMI observations add additional DOFS to the GOSAT posterior at 0°–30° N (mainly over India and the Middle East, **Figure 5 and Figure S3**), where TROPOMI has more observations than in the rest of the world (Figure 1)."

L324: Is this the same for GOSAT?

We added the following sentence to this paragraph:

**"The GOSAT-only inversion without error weighting for wetlands shows no such problems, but we still apply error weighting in that base inversion for comparison to TROPOMI."**

L347: Why does the GOSAT inversion show negative biases against the GOSAT observations almost everywhere? Is that contributed by the strong prior constraint to wetland emissions?

We added the following sentence after the cited one:

**"GOSAT observations are still underestimated by an average of 5.3 ppbv in the GOSAT inversion because the information from the observations is not sufficient to fully correct the bias in the prior estimate."**

L371-372: Can you estimate magnitude of the potential errors due to the strong constraint on wetland emissions?

We changed the sentence to:

"Our restricted adjustment of wetland emissions means that errors in wetland emissions could be projected to non-wetland sectors. **For example, for the TROPOMI-only inversion, global posterior non-wetland emissions are 361 Tg a$^{-1}$ in the base inversion and 389 Tg a$^{-1}$ in the sensitivity inversion without increased weight for wetland emissions (Table S1). For the GOSAT inversion the effect is much less, 399 versus 404 Tg a $^{-1}$ (Table S2)."**

L386: Specify the version of EDGAR.

We changed the sentence to:

"This has been a long-standing result of inversions of satellite data using EDGAR **v4.1 and v4.2** as prior estimate … but GFEI takes its spatial allocation of coal emissions from EDGAR **v4.3.2**."

L392: Not clear what is "inconsistent" with the TROPOMI inversion.

We changed the sentence to:

"Our TROPOMI inversion over Southeast China shows spatially inconsistent results **with the GOSAT inversion** (Figure 5) and overcorrects …"

L402: What is the "base" inversion?

We made the following modifications to 7$^{th}$ paragraph of Section 3.2:

"**In addition to the base inversion as described above,** we examined the sensitivity to the choice of $\gamma$ with sensitivity inversions…"

L424: The number of 34 Tg a$^{-1}$ seems not consistent with Fig. 7, which looks like around 40 Tg a$^{-1}$. Furthermore, I am not sure why the joint inversion increased the emissions estimate, as each inversion showed downward estimates from the prior estimate.

A joint inversion is not a linear addition of the adjustments from the TROPOMI and GOSAT inversions. In the joint inversion, the DOFS in the south-central US, where methane emissions show large upward adjustments, becomes relatively large compared to the rest of the US. The large upward adjustments in the south-central US therefore contribute to the upward adjustments in the US posterior emissions.

We changed the last sentence in Section 4.3.2 to:

"The joint inversion adjusts emissions upwards to **40** Tg a$^{-1}$ due to the larger **averaging kernel** sensitivity over the south-central US, **where emissions have large upward adjustments**."

We also update the national emissions in Section 4.3.2 to be consistent with Figure 7.